# How the level sampling process impacts zero-shot generalisation in deep reinforcement learning

## Abstract

A key limitation preventing the wider adoption of autonomous agents trained via deep reinforcement learning (RL) is their limited ability to generalise to new environments, even when these share similar characteristics with environments encountered during training. In this work, we investigate how a non-uniform sampling strategy of individual environment instances, or levels, affects the zero-shot generalisation (ZSG) ability of RL agents, considering two failure modes: overfitting and over-generalisation. As a first step, we measure the mutual information (MI) between the agent's internal representation and the set of training levels, which we find to be well-correlated to instance overfitting. In contrast to uniform sampling, adaptive sampling strategies prioritising levels based on their value loss are more effective at maintaining lower MI, which provides a novel theoretical justification for this class of techniques. We then turn our attention to unsupervised environment design (UED) methods, which adaptively *generate* new training levels and minimise MI more effectively than methods sampling from a fixed set. However, we find UED methods significantly *shift* the training distribution, resulting in over-generalisation and worse ZSG performance over the distribution of interest. To prevent both instance overfitting and over-generalisation, we introduce *self-supervised environment design* (SSED). SSED generates levels using a variational autoencoder, effectively reducing MI while minimising the shift with the distribution of interest, and leads to statistically significant improvements in ZSG over fixed-set level sampling strategies and UED methods.

## 1 Introduction

A central challenge facing modern reinforcement learning (RL) is learning policies capable of zero-shot transfer of learned behaviours to a wide range of environment settings. Prior applications of RL algorithms (Agostinelli et al., 2019; Lee et al., 2020; Rudin et al., 2021) indicate that strong zero-shot generalisation (ZSG) can be achieved through an adaptive sampling strategy over the set of environment instances available during training, which we refer to as the set of training *levels*. However the relationship between ZSG and the level sampling process remains poorly understood. In this work,

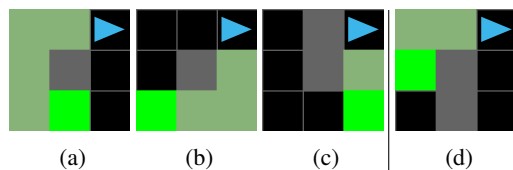

|     |     |     |     |
|:---:|:---:|:---:|:---:|
| (a) | (b) | (c) | (d) |

Figure 1: The agent (blue) must navigate to the goal (lime green) but cannot pass through walls (grey) and only observes tiles directly adjacent to itself. An agent trained over levels (a)-(c) will transfer zero-shot to level (d) if it has learnt a behavior adapted to the task semantics of following pale green tiles to reach the goal location.

we draw novel connections between this process and the minimisation of an upper bound on the generalisation error derived by Bertran et al. (2020), which depends on the *mutual information* (MI) between the agent's internal representation and the identity of individual training levels.

An agent learning level-specific policies implies high MI between its internal representation and the level identities, and, in general, will not transfer zero-shot to new levels. To build an understanding of the relationship between MI and ZSG, consider the minimal gridworld navigation example in Figure 1. A "shortcut" exists in level (a), and a model with high MI is able to first predict the level identity from its initial observation to learn an ensemble of level-specific policy optimal over the training set. When deployed on (d) the model will predict it is in (a) since under the agent's restricted

field of view (a) and (d) share the same initial observation. As a result the agent will attempt to follow the (a)-specific policy, which will not transfer. We discover that the reduced generalisation error achieved by adaptive level sampling strategies over uniform sampling can be attributed to their effectiveness in reducing the MI between the agent's internal representation and the level identity. In particular, we find that strategies de-prioritising levels with low value loss, as proposed in prioritised level replay (PLR, Jiang et al., 2021b), implicitly minimise mutual information as they avoid training on levels in which the value function is accurately estimated through level identification.

While some adaptive sampling strategies reduce the generalisation gap, their effectiveness is ultimately limited by the number of training levels. We propose *Self-Supervised Environment Design* (SSED) which augments the set of training levels to further reduce generalisation error. We find training on an augmented set can negatively impact performance when the augmented set is not drawn from the same distribution as the training set. We show it induces a form of *over-generalisation*, in which the agent learns to solve levels incompatible with the targeted task, and performs poorly at test time. There is therefore a trade-off between augmenting the training set to prevent instance-overfitting, i.e. to avoid learning level-specific policies, and ensuring that this augmented set comes from the same distribution to avoid distributional shift and over-generalisation. In our experiments, we show that SSED strikes this trade-off more effectively than other adaptive sampling and environment design methods. SSED achieves significant improvements in the agent's ZSG capabilities, reaching 1.25 times the returns of the next best baseline on held-out levels, and improving performance by two to three times on more difficult instantiations of the target task.

## 2 RELATED WORK

**Buffer-free sampling strategies.** Domain randomisation (DR, Tobin et al., 2017; Jakobi, 1997), is one of the earliest proposed methods for improving the generalisation ability of RL agents by augmenting the set of available training levels, and does so by sampling uniformly between manually specified ranges of environment parameters. Subsequent contributions introduce an implicit prioritisation over the generated set by inducing a minimax return (robust adversarial RL Pinto et al., 2017) or a minimax regret (unsupervised environment design (UED), Dennis et al., 2020) game between the agent and a *level generator*, which are trained concurrently. These adversarial formulations prioritise levels in which the agent is currently performing poorly to encourage robust generalisation over the sampled set, with UED achieving better Nash equilibrium theoretical guarantees. CLUTR (Azad et al., 2023) removes the need for domain-specific RL environments and improves sample efficiency by having the level generator operate within a low dimensional latent space of a generative model pre-trained on randomly sampled level parameters. However the performance and sample efficiency of these methods is poor when compared to a well calibrated DR implementation or to the buffer-based sampling strategies discussed next.

**Buffer-based sampling strategies.** Prioritised sampling is often applied to off-policy algorithms, where individual transitions in the replay buffer are prioritised (Schaul et al., 2015) or resampled with different goals in multi-goal RL (Andrychowicz et al., 2017; Zhang et al., 2020). Prioritised Level Replay (PLR, Jiang et al., 2021b) instead affects the sampling process of *future* experiences, and is thus applicable to both on- and off-policy algorithms. PLR maintains a buffer of training levels and empirically demonstrates that prioritising levels using a scoring function proportional to high value prediction loss results in better sample efficiency and improved ZSG performance. Robust PLR (RPLR, Jiang et al., 2021a) extends PLR to the UED setting, using DR as its level generation mechanism, whereas ACCEL (Parker-Holder et al., 2022) gradually evolves new levels by performing random edits on high scoring levels in the buffer. SAMPLR (Jiang et al., 2022) proposes to eliminate the covariate shift induced by the prioritisation strategy by modifying *individual transitions* using a second simulator that runs in parallel. However SAMPLR is only applicable to settings in which the level parameter distribution is provided, whereas SSED can approximate this distribution from a dataset of examples.

**Mutual-information minimisation in RL.** In prior work, mutual information has been minimised in order to mitigate instance-overfitting, either by learning an ensemble of policies (Bertran et al., 2020; Ghosh et al., 2021), performing data augmentation on observations (Raileanu et al., 2021; Kostrikov et al., 2021), an auxiliary objective (Dunion et al., 2023) or introducing information bottlenecks through selective noise injection on the agent model (Igl et al., 2019; Cobbe et al., 2019).

In contrast, our work is the first to draw connections between mutual-information minimisation and adaptive level sampling and generation strategies.

## 3 PRELIMINARIES

**Reinforcement learning.** We model an individual level as a Partially Observable Markov Decision Process (POMDP) $\langle A, O, S, \mathcal{T}, \Omega, R, p_0, \gamma \rangle$ where $A$ is the action space, $O$ is the observation space, $S$ is the set of states, $\mathcal{T} : S \times A \to \Delta(S)$ and $\Omega : S \to \Delta(O)$ are the transition and observation functions (we use $\Delta(\cdot)$ to indicate these functions map to distributions), $R : S \to \mathbb{R}$ is the reward function, $p_0(\mathrm{s})$ is the initial state distribution and $\gamma$ is the discount factor. We consider the episodic RL setting, in which the agent attempts to learn a policy $\pi$ maximising the expected discounted return $V^\pi(s_t) = \mathbb{E}_\pi[\sum_{\bar{t}=t}^T \gamma^{t-\bar{t}} r_t]$ over an episode terminating at timestep $T$, where $s_t$ and $r_t$ are the state and reward at step $t$. We use $V^\pi$ to refer to $V^\pi(s_0)$, the expected episodic returns taken from the first timestep of the episode. In this work, we focus on on-policy actor-critic algorithms (Mnih et al., 2016; Lillicrap et al., 2016; Schulman et al., 2017) representing the agent policy $\pi_{\boldsymbol{\theta}}$ and value estimate $\hat{V}_{\boldsymbol{\theta}}$ with neural networks (in this paper we use $\boldsymbol{\theta}$ to refer to model weights). The policy and value networks usually share an intermediate state representation $b_{\boldsymbol{\theta}}(o_t)$ (or for recurrent architectures $b_{\boldsymbol{\theta}}(H_t^o)$, $H_t^o = \{o_0, \cdots, o_t\}$ being the history of observations $o_i$).

**Contextual MDPs.** Following Kirk et al. (2023), we model the set of environment instances we aim to generalise over as a Contextual-MDP (CMDP) $\mathcal{M} = \langle A, O, S, \mathcal{T}, \Omega, R, p_0(\mathrm{s}|\mathbf{x}), \gamma, X_C, p(\mathbf{x}) \rangle$. The CMDP may be viewed as a POMDP, except that the reward, transition and observation functions now also depend on the *context set* $X_C$ with associated distribution $p(\mathbf{x})$, that is $\mathcal{T} : S \times X_C \times A \to \Delta(S)$, $\Omega : S \times X_C \to \Delta(O)$, $R : S \times X_C \to \mathbb{R}$. Each element $\boldsymbol{x} \in X_C$ is not observable by the agent and instantiates a *level* $i_{\boldsymbol{x}}$ of the CMDP with initial state distribution $p_0(\mathrm{s}|\mathbf{x})$. The optimal policy of the CMDP maximises $V_C^\pi = \mathbb{E}_{\mathbf{x} \sim p(\mathbf{x})}[V_{i_{\boldsymbol{x}}}^\pi]$, with $V_{i_{\boldsymbol{x}}}^\pi$ referring to the expected return in level $i_{\boldsymbol{x}}$ instantiated by $\boldsymbol{x}$ (we use $L$ to refer to a set of levels $i$ and $X$ to refer to a set of level parameters $\boldsymbol{x}$). We assume access to a parametrisable simulator with parameter space $\mathbb{X}$, with $X_C \subset \mathbb{X}$. While prior work expects $X_C$ to correspond to all solvable levels in $\mathbb{X}$, we consider the more general setting in which there may be more than one CMDP within $\mathbb{X}$, whereas we aim to solve a specific target CMDP. We refer to levels with parameters $\boldsymbol{x} \in X_C$ as *in-context* and to levels with parameters $\boldsymbol{x} \in \mathbb{X} \setminus X_C$ as *out-of-context*. As we show in our experiments, training on out-of-context levels can induce distributional shift and cause the agent to learn a different policy than the optimal CMDP policy.

**Generalisation bounds.** We start training with access to a limited set of level parameters $X_{\text{train}} \subset X_C$ sampled from $p(\mathbf{x})$, and evaluate generalisation using a set of held-out level parameters $X_{\text{test}}$, also sampled from $p(\mathbf{x})$. Using the unbiased value estimator lemma from Bertran et al. (2020),

**Lemma 3.1** *Given a policy $\pi$ and a set $L = \{i_{\boldsymbol{x}} | \boldsymbol{x} \sim p(\mathbf{x})\}_n$ of $n$ levels from a CMDP with context distribution $p(\mathbf{x})$, we have $\forall H_t^o$ ($t < \infty$) compatible with $L$ (that is the observation sequence $H_t^o$ occurs in $L$), $\mathbb{E}_{L|H_t^o}[V_{i_{\boldsymbol{x}}}^\pi(H_t^o)] = V_C^\pi(H_t^o)$, with $V_{i_{\boldsymbol{x}}}^\pi(H_t^o)$ being the expected returns under $\pi$ given observation history $H_t^o$ in a given level $i_{\boldsymbol{x}}$, and $V_C^\pi(H_t^o)$ being the expected returns across all possible occurrences of $H_t^o$ in the CMDP.*

we can estimate the generalisation gap using a formulation reminiscent of supervised learning,

$$\text{GenGap}(\pi) := \frac{1}{|X_{\text{train}}|} \sum_{\boldsymbol{x} \in X_{\text{train}}} V_{i_{\boldsymbol{x}}}^\pi - \frac{1}{|X_{\text{test}}|} \sum_{\boldsymbol{x} \in X_{\text{test}}} V_{i_{\boldsymbol{x}}}^\pi. \tag{1}$$

Using this formulation, Bertran et al. (2020) extend generalisation results in the supervised setting (Xu & Raginsky, 2017) to derive an upper bound for the GenGap.

**Theorem 3.2** *For any CMDP such that $|V_C^\pi(H_t^o)| \le D/2, \forall H_t^o, \pi$, then for any set of training levels $L$, and policy $\pi$*

$$GenGap(\pi) \le \sqrt{\frac{2D^2}{|L|} \times MI(L, \pi)}. \tag{2}$$

With $\text{MI}(L, \pi) = \sum_{i \in L} \text{MI}(i, \pi)$ being the mutual information between $\pi$ and the identity of each level $i \in L$. In this work, we show that minimising the bound in Theorem 3.2 is an effective surrogate objective for reducing the GenGap.

**Adaptive level sampling.** We study the connection between $\text{MI}(L, \pi)$ and adaptive sampling strategies over $L$. PLR introduce a scoring function $\textbf{score}(\tau_i, \pi)$ compute level scores from a rollout trajectory $\tau_i$. Scores are used to define an adaptive sampling distribution over a level buffer $\Lambda$, with

$$P_\Lambda = (1 - \rho) \cdot P_S + \rho \cdot P_R, \tag{3}$$

where $P_S$ is a distribution parametrised by the level scores and $\rho$ is a coefficient mixing $P_S$ with a staleness distribution $P_R$ that promotes levels replayed less recently. Jiang et al. (2021b) experiment with different scoring functions, and empirically find that the scoring function based on the $\ell_1$-value loss $S_i^V = \textbf{score}(\tau_i, \pi) = (1/|\tau_i|) \sum_{H_t^o \in \tau_i} |\hat{V}(H_t^o) - V_i^\pi(H_t^o)|$ incurs a significant reduction in the GenGap at test time.

In the remaining sections, we draw novel connections between the $\ell_1$-value loss prioritisation strategy and the minimisation of $\text{MI}(L, \pi)$. We then introduce SSED, a level generation and sampling framework training the agent over an augmented set of levels. SSED jointly minimises $\text{MI}(L, \pi)$ while increasing $|L|$ and as such is more effective at minimising the bound from Theorem 3.2.

## 4    MUTUAL-INFORMATION MINIMISATION UNDER A FIXED SET OF LEVELS

We begin by considering the setting in which $L$ remains fixed. We make the following arguments: 1) as the contribution of each level to $\text{MI}(L, \pi)$ is generally *not uniform* across $L$ nor *constant* over the course of training, an adaptive level sampling strategy yielding training data with low $\text{MI}(L, \pi)$ can reduce the GenGap over uniform sampling; 2) the value prediction objective promotes learning internal representations informative of the current level identity and causes overfitting; 3) deprioritising levels with small value loss implicitly reshapes the training data distribution to yield smaller $\text{MI}(L, \pi)$, reducing GenGap. We substantiate our arguments with a comparison of different sampling strategies in the Procgen benchmark (Cobbe et al., 2020).

### 4.1    MAINTAINING LOW MUTUAL INFORMATION CONTENT VIA ADAPTIVE SAMPLING

The following lemma enables us to derive an upper bound for $\text{MI}(L, \pi)$ that can be approximated using the activations of the state representation shared between the actor and critic networks.

**Lemma 4.1** *(proof in appendix) Given a set of training levels $L$ and an agent model $\pi = f \circ b$, where $b(H_t^o) = h_t$ is an intermediate state representation and $f$ is the policy head, we can bound $MI(L, \pi \circ b)$ by $MI(L, b)$, which in turn satisfies*

$$MI(L, \pi) \leq MI(L, b) = \mathcal{H}(p(\text{i})) + \sum_{i \in L} \int dh\, p(\textbf{h}, \text{i}) \log p(\text{i}|\textbf{h}) \tag{4}$$

$$\approx \mathcal{H}(p(\text{i})) + \frac{1}{|B|} \sum_{(i, H_t^o) \in B} \log p(\text{i}|b(H_t^o)) \tag{5}$$

*where $\mathcal{H}(p)$ is the entropy of $p$ and $B$ is a batch of trajectories collected from levels $i \in L$.*

This result applies to any state representation function $b$, including the non-recurrent case where $b(H_t^o) = b(o_t), \forall (o, H_t^o) \in (O, O^{\otimes t})$. To remain consistent with the CMDP we must set $p(\text{i})$ to $p(\textbf{x})$, making the entropy $\mathcal{H}(p(\text{i}))$ a constant. However the second term in Equation (5) depends on the representation $b$ learned under the training data. We hypothesise that minimising $\text{MI}(L, b)$ in the training data is an effective data regularisation technique against instance-overfitting. We can isolate level-specific contributions to $\text{MI}(L, b)$ as

$$\sum_{(i, H_t^o) \in B} \log p(\text{i}|b(H_t^o)) = \sum_{i \in L} \sum_{H_t^o \in B_i} \log p(\text{i}|b(H_t^o)), \tag{6}$$

where $B_i$ indicates the batch trajectories collected from level $i$. As sampled trajectories depend on the behavioral policy, and as the information being retained depends on $b$, each level's contribution to $\text{MI}(L, b)$ is in general not constant over the course of training nor uniform across $L$. There should therefore exist adaptive distributions minimising $\text{MI}(L, b)$ more effectively than uniform sampling.

### 4.2    ON THE EFFECTIVENESS OF VALUE LOSS PRIORITISATION STRATEGIES

From a representation learning perspective, the value prediction objective may be viewed as a self-supervised auxiliary objective shaping the intermediate state representation $b$. This additional signal

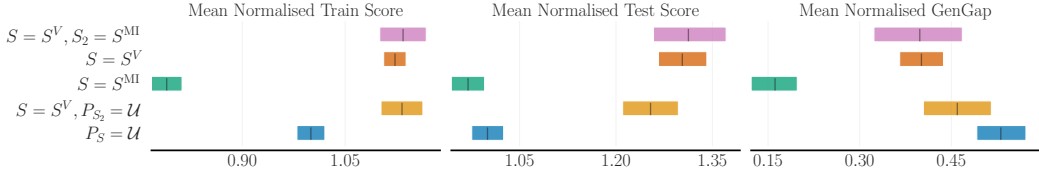

Figure 2: Mean aggregated train and test scores and GenGap of different sampling strategies. Scores are normalised for each game using the mean score achieved by the uniform sampling strategy (we use the test set scores to normalise GenGap).

is often necessary for learning, and motivates sharing $b$ across the policy and value networks. However, in a CMDP the value prediction loss

$$L_V(\boldsymbol{\theta}) = \frac{1}{|B|} \sum_{(i,H_t^o) \in B} (\hat{V}_{\boldsymbol{\theta}}(H_t^o) - V_i^\pi(H_t^o))^2 \qquad (7)$$

uses level-specific functions $V_i^\pi$ as targets, which may be expressed as $V_i^\pi = V_C^\pi + v_i^\pi$, where $v_i^\pi$ is a component specific to $i$. While Lemma 3.1 guarantees convergence to an unbiased estimator for $V_C^\pi$ when minimising Equation (7), reaching zero training loss is only achievable by learning the level specific components $v_i^\pi$. Perfect value prediction requires learning an intermediate representation from which the current level $i$ is identifiable, which implies high $\mathrm{MI}(i,b)$. Conversely, we can characterise PLR's $\ell_1$-value loss sampling as a data regularisation technique minimising $\mathrm{MI}(L,b)$ when generating the training data. By de-prioritising levels with low $L_{V_i}$, PLR prevents the agent from generating training data for which its internal representation has started overfitting to.

### 4.3 COMPARING MUTUAL INFORMATION MINIMISATION SAMPLING STRATEGIES

We aim to establish how PLR under value loss scoring $S^V$ compares to a scoring function based on Equation (6). We define this function as $S_i^{\mathrm{MI}} = \sum_{t=0}^{T} \log p_{\boldsymbol{\theta}}(\mathrm{i}|b(H_t^o))$, where $p_{\boldsymbol{\theta}}$ is a linear classifier. We also introduce a secondary scoring strategy $S_2$ with associated distribution $P_{S_2}$, letting us *mix* different sampling strategies and study their interaction. $P_\Lambda$ then becomes

$$P_\Lambda = (1 - \rho) \cdot ((1 - \eta) \cdot P_S + \eta \cdot P_{S_2}) + \rho \cdot P_R, \qquad (8)$$

with $\eta$ being a mixing parameter. We compare different sampling strategies in Procgen, a benchmark of 16 games designed to measure generalisation in RL. We train the PPO (Schulman et al., 2017) baseline employed in (Cobbe et al., 2020), which uses a non-recurrent intermediate representation $b_{\boldsymbol{\theta}}(o_t)$ in the "easy" setting ($|L| = 200$, $25M$ timesteps). We report a complete description of the experimental setup in Appendix D.1.

Figure 2, compares value loss scoring ($S = S^V$), uniform sampling ($P_S = \mathcal{U}$), $\mathcal{U}(\cdot)$ being the uniform distribution, direct MI minimisation ($S = S^{\mathrm{MI}}$) as well as mixed strategies ($S = S^V, S_2 = S^{\mathrm{MI}}$) and ($S = S^V, P_{S_2} = \mathcal{U}$). While ($S = S^{\mathrm{MI}}$) reduces GenGap the most, the degradation it induces in the training performance outweigh its regularisation benefits. This result is consistent with Theorem 3.2 and Lemma 4.1, as $\mathrm{MI}(L,b)$ bounds the GenGap and not the test returns.[1] On the other-hand, ($S = S^V$) slightly improves training efficiency while reducing the GenGap. As denoted by its smaller GenGap when compared to ($P_S = \mathcal{U}$), the improvements achieved by ($S = S^V$) are markedly stronger over the test set than for the train set, and indicate that the main driver behind the stronger generalisation performance is not a higher sample efficiency but a stronger regularisation. We tested different mixed strategies ($S = S^V, S_2 = S^{\mathrm{MI}}$) using different $\eta$, and the best performing configuration (reported in Figure 2) only achieves a marginal improvement over ($S = S^V$) (on the other hand, mixing $S^V$ and uniform sampling ($S = S^V, P_{S_2} = \mathcal{U}$) noticeably reduces the test set performance). This implies that ($S = S^V$) strikes a good balance between training efficiency and regularisation within the space of mutual information minimisation adaptive sampling strategies. In Appendix B.1 we analyse the correlation between $\mathrm{MI}(L,b)$, the $\ell_1$-value loss and the GenGap across all procgen games and methods tested. We find $\mathrm{MI}(L,b)$ to be positively correlated to the GenGap ($p < 1\mathrm{e}{-34}$) and inversely correlated with the $\ell_1$-value loss ($p < 1\mathrm{e}{-16}$).

---

[1]Exclusively focusing on data regularisation can be problematic: in the most extreme case, destroying all information contained within the training data would guarantee $\mathrm{MI}(L,\pi) = \mathrm{GenGap} = 0$ but it would also make the performance on the train and test sets equally bad.

## 5 SELF-SUPERVISED ENVIRONMENT DESIGN

We have established that certain adaptive sampling strategies effectively minimise $\text{MI}(L, b)$, which in turn reduces GenGap. However our experiments in Section 4 and appendices B.1 and B.2 indicate GenGap may still be significant when training the agent over a fixed level set, even with an adaptive sampling strategy. We now introduce SSED, a framework designed to more aggressively minimise the generalisation bound in Theorem 3.2 by jointly minimising $\text{MI}(L, b)$ and increasing $|L|$. SSED does so by generating an augmented set of training levels $\tilde{L} \supset L$, while still employing an adaptive sampling strategy over the augmented set.

SSED shares UED's requirement of having access to a parametrisable simulator allowing the specification of levels through environment parameters $\boldsymbol{x}$. In addition, we assume that we start with a limited set of level parameters $X_{\text{train}}$ sampled from $p(\mathbf{x})$. SSED consists of two components: a *generative phase*, in an augmented set $\tilde{X}$ is generated using a batch $X \sim \mathcal{U}(X_{\text{train}})$ and added to the buffer $\Lambda$, and a *replay phase*, in which we use the adaptive distribution $P_{\Lambda}$ to sample levels from $\Lambda$. We alternate between the generative and replay phases, and only perform gradient updates on the agent during the replay phase. Algorithm 1 describes the full SSED pipeline, and we provide further details on each phase below.

---

**Algorithm 1** Self-Supervised Environment Design

---

**Input:** Pre-trained VAE encoder and decoder networks $\psi_{\boldsymbol{\theta}_E}, \phi_{\boldsymbol{\theta}_D}$, level parameters $X_{\text{train}}$, number of pairs $M$, number of interpolations per pair $K$

1: Initialise agent policy $\pi$ and level buffer $\Lambda$, adding level parameters in $X_{\text{train}}$ to $\Lambda$
2: Update $X_{\text{train}}$ with variational parameters $(\boldsymbol{\mu}_{\mathbf{z}}, \boldsymbol{\sigma}_{\mathbf{z}})_n \leftarrow \psi_{\boldsymbol{\theta}_E}(\boldsymbol{x}_n)$ **for** $\boldsymbol{x}_n$ in $X_{\text{train}}$
3: **while** *not converged* **do**
4:     Sample batch $X$ using $P_{\Lambda}$                                      ▷ Replay phase
5:     **for** $\boldsymbol{x}$ in $X$ **do**
6:        Collect rollouts $\tau$ from $i_{\boldsymbol{x}}$ and compute scores $S, S_2$
7:        Update $\pi$ according to $\tau$
8:        Update scores $S, S_2$ of $\boldsymbol{x}$ in $\Lambda$
9:     Randomly sample $2M$ $(\boldsymbol{x}, \boldsymbol{\mu}, \boldsymbol{\sigma})$ from $X_{\text{train}}$ and arrange them into $M$ pairs.
10:    **for** $((\boldsymbol{x}, \boldsymbol{\mu}_{\mathbf{z}}, \boldsymbol{\sigma}_{\mathbf{z}})_i, (\boldsymbol{x}, \boldsymbol{\mu}_{\mathbf{z}}, \boldsymbol{\sigma}_{\mathbf{z}})_j)$ in pairs **do**             ▷ Generative phase
11:     Compute $K$ interpolations $\{(\boldsymbol{\mu}_{\mathbf{z}}, \boldsymbol{\sigma}_{\mathbf{z}})\}_K$ between $((\boldsymbol{x}, \boldsymbol{\mu}_{\mathbf{z}}, \boldsymbol{\sigma}_{\mathbf{z}})_i, (\boldsymbol{x}, \boldsymbol{\mu}_{\mathbf{z}}, \boldsymbol{\sigma}_{\mathbf{z}})_j)$
12:     **for** $(\boldsymbol{\mu}_{\mathbf{z}}, \boldsymbol{\sigma}_{\mathbf{z}})_k$ in $\{(\boldsymbol{\mu}_{\mathbf{z}}, \boldsymbol{\sigma}_{\mathbf{z}})\}_K$ **do**
13:        Sample embedding $\boldsymbol{z} \sim \mathcal{N}(\boldsymbol{\mu}_{\mathbf{z}}, \boldsymbol{\sigma}_{\mathbf{z}})$
14:        $\tilde{\boldsymbol{x}} \leftarrow \phi_{\boldsymbol{\theta}_D}(\boldsymbol{z})$
15:        Collect $\pi$'s trajectory $\tau$ from $\boldsymbol{x}$ and compute scores $S, S_2$
16:        Add $\langle \boldsymbol{x}, S, S_2 \rangle$ to $\Lambda$ at $\arg\min_{\{\Lambda \backslash X_{\text{train}}\}} S_2$ **if** $S_2 > \min_{\{\Lambda \backslash X_{\text{train}}\}} S_2$

---

### 5.1 THE GENERATIVE PHASE

While SSED is not restricted to a particular approach to obtain $\tilde{X}$, we chose the VAE (Kingma & Welling, 2014; Rezende et al., 2014) due its ability to model the underlying training data distribution $p(\mathbf{x})$ as stochastic realisations of a latent distribution $p(\mathbf{z})$ via a generative model $p(\mathbf{x} \mid \mathbf{z})$. The model is pre-trained on $X_{\text{train}}$ by maximising the variational ELBO

$$\mathcal{L}_{\text{ELBO}} = \mathbb{E}_{\boldsymbol{x} \sim p(\mathbf{x})} \left\{ \mathbb{E}_{\boldsymbol{z} \sim q(\mathbf{z}|\mathbf{x}; \psi_{\boldsymbol{\theta}_E})} [\log p(\mathbf{x} \mid \mathbf{z}; \phi_{\boldsymbol{\theta}_D})] - \beta D_{\text{KL}}(q(\mathbf{z} \mid \mathbf{x}; \psi_{\boldsymbol{\theta}_E}) \,||\, p(\mathbf{z})) \right\}, \quad (9)$$

where $q(\mathbf{z} \mid \mathbf{x}; \psi_{\boldsymbol{\theta}_E})$ is a variational approximation of an intractable model posterior distribution $p(\mathbf{z} \mid \mathbf{x})$ and $D_{\text{KL}}(\cdot \,||\, \cdot)$ denotes the Kullback–Leibler divergence, which is balanced using the coefficient $\beta$, as proposed by Higgins et al. (2017). The generative $p(\mathbf{x} \mid \mathbf{z}; \phi_{\boldsymbol{\theta}_D})$ and variational $q(\mathbf{z} \mid \mathbf{x}; \psi_{\boldsymbol{\theta}_E})$ models are parametrised via encoder and decoder networks $\psi_{\boldsymbol{\theta}_E}$ and $\phi_{\boldsymbol{\theta}_D}$.

We use $p(\mathbf{x}; \phi_{\boldsymbol{\theta}_D})$ to generate augmented level parameters $\tilde{\boldsymbol{x}}$. As maximising Equation (9) fits the VAE such that the marginal $p(\mathbf{x}; \phi_{\boldsymbol{\theta}_D}) = \int p(\mathbf{x} \mid \mathbf{z}; \phi_{\boldsymbol{\theta}_D}) p(\mathbf{z}) \, d\mathbf{z}$ approximates the data distribution $p(\mathbf{x})$, and sampling from it limits distributional shift. This makes out-of-context levels less frequent, and we show in Section 6 that this aspect is key in enabling SSED-trained agents to outperform UED-agents. To improve the quality of the generated $\tilde{\boldsymbol{x}}$, we interpolate in the latent space between

the latent representations of pair of samples $(\boldsymbol{x}_i, \boldsymbol{x}_j) \sim X_{\text{train}}$ to obtain $\mathbf{z}$, instead of sampling from $p(\mathbf{z})$, as proposed by White (2016). We evaluate the agent (without updating its weights) on the levels obtained from a batch of levels parameters $\tilde{X}$, adding to the buffer $\Lambda$ any level scoring higher than the lowest scoring generated level in $\Lambda$. We provide additional details on the architecture, hyperparameters and pre-training process in Appendix D.3.

## 5.2 THE REPLAY PHASE

All levels in $X_{\text{train}}$ originate from $p(\mathbf{x})$ and are in-context, whereas generated levels, being obtained from an approximation of $p(\mathbf{x})$, do not benefit from as strong of a guarantee. As training on out-of-context levels can significantly harm the agents' performance on the CMDP, we control the ratio between $X_{\text{train}}$ and augmented levels using Equation (8) to define $P_\Lambda$. $P_S$ and $P_R$ only sample from $X_{\text{train}}$ levels, whereas $P_{S_2}$ supports the entire buffer. We set both $S_1$ and $S_2$ to score levels according to the $\ell_1$-value loss. We initialise the buffer $\Lambda$ to contain $X_{\text{train}}$ levels and gradually add generative phase levels over the course of training. A level gets added if $\Lambda$ is not full or if it scores higher than the lowest scoring level in the buffer. We only consider levels solved at least once during generative phase rollouts to ensure unsolvable levels do not get added in. We find out-of-context levels to be particularly harmful in the early stages of training, and reduce their frequency early on by linearly increasing the mixing parameter $\eta$ from 0 to 1 over the course of training.

## 6 EXPERIMENTS

As it only permits level instantiation via providing a random seed, Procgen's level generation process is both uncontrollable and unobservable. Considering the seed space to be the level parameter space $\mathbb{X}$ makes the level generation problem trivial as it is only possible to define a single CMDP with $X_C = \mathbb{X}$ and $p(\mathbf{x}) = \mathcal{U}(\mathbb{X})$. Instead, we wish to demonstrate SSED's capability in settings where $X_C$ spans a (non-trivial) manifold in $\mathbb{X}$, i.e. only specific parameter semantics will yield levels of the CMDP of interest. As such we pick Minigrid, a partially observable gridworld navigation domain (Chevalier-Boisvert et al., 2018). Minigrid levels can be instantiated via a parameter vector describing the locations, starting states and appearance of the objects in the grid. Despite its simplicity, Minigrid qualifies as a parametrisable simulator capable of instantiating multiple CMDPs. We define the context space of our target CMDP as spanning the layouts where the location of green "moss" tiles and orange "lava" tiles are respectively positively and negatively correlated to their distance to the goal location. We employ procedural generation to obtain a set $X_{\text{train}}$ of 512 level parameters, referring the reader to Figure 10 for a visualisation of levels from $X_{\text{train}}$, and to Appendix C for extended details on the CMDP specification and level set generation process.

As the agent only observes its immediate surroundings and does not know the goal location a priori, the optimal CMDP policy is one that exploits the semantics shared by all levels in the CMDP, exploring first areas with high perceived moss density and avoiding areas with high lava density. Our CMDP coexists alongside a multitude of other potential CMDPs in the level space and some correspond to incompatible optimal policies (for example levels in which the correlation of moss and lava tiles with the goal is reversed). As such, it is important to maintain consistency with the CMDP semantics when generating new levels.

We compare SSED to multiple baselines, sorted in two sets. The first set of baselines is restricted to sample from $X_{\text{train}}$, and consists of uniform sampling ($\mathcal{U}$) and PLR with the $\ell_1$-value loss strategy. The second set incorporates a generative mechanism and as such are more similar to SSED. We consider domain randomisation (DR) (Tobin et al., 2017) which generates levels by sampling uniformly between pre-determined ranges of parameters, RPLR (Jiang et al., 2021a), which combines PLR with DR used as its generator, and the current UED state-of-the-art, ACCEL Parker-Holder et al. (2022), an extension of RPLR replacing DR by a generator making local edits to currently high scoring levels in the buffer. All experiments share the same PPO (Schulman et al., 2017) agent, which uses the LSTM-based architecture and hyperparameters reported in Parker-Holder et al. (2022), training over 27k updates.

### 6.1 GENERALISATION PERFORMANCE

As shown in Figure 3, SSED achieves statistically significant improvements in its IQM (inter-quantile mean), mean score, optimality gap and mean solved rate over other methods on held-out levels from the CMDP. SSED's ZSG performance on held-out levels from $X_{\text{train}}$ demonstrates it al-

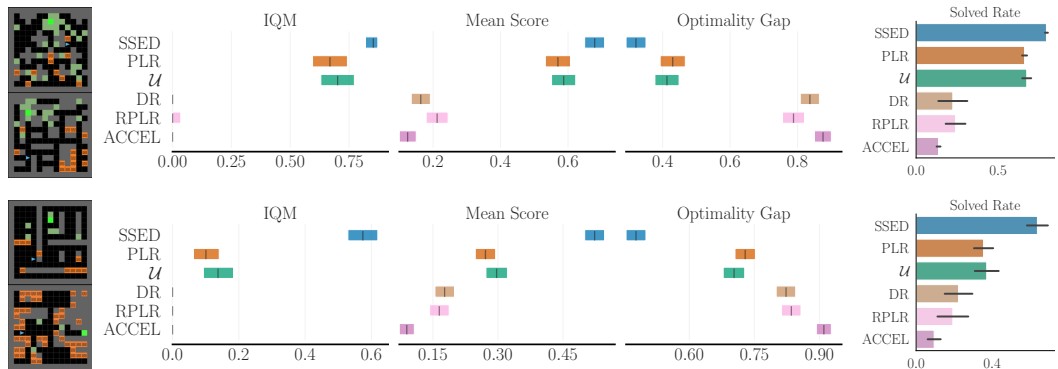

Figure 3: Center: aggregate test performance on 200 held-out levels from $X_{\text{train}}$ (top) and in-context edge cases (bottom). Right: zero-shot solved rate on the same levels, the bars indicate standard error for 3 training seeds (some example levels are provided for reference, refer to Appendix C for additional details on our evaluation sets).

leviates instance-overfitting while remaining consistent with the target CMDP. This is thanks to its generative model effectively approximating $p(\mathbf{x})$, and to its mixed sampling strategy ensuring many training levels originate from $X_{\text{train}}$, which are guaranteed to be in-context. We next investigate whether SSED's level generation improves robustness to *edge cases* which are in-context but would have a near zero likelihood of being in $X_{\text{train}}$ in a realistic setting. We model edge cases as levels matching the CMDP semantics but generated via different procedural generation parameters. We find SSED to be particularly dominant is this setting, achieving a solved rate and IQM respectively two- and four-times $\mathcal{U}$, the next best method, and a mean score 1.6 times PLR, the next best method for that metric. SSED is therefore capable of introducing additional diversity in the level set in a manner that remains semantically consistent with the CMDP. In Figure 4, we measure transfer to levels of increased complexity using a set of layouts 9 times larger in area than $X_{\text{train}}$ levels and which would be impossible to instantiate during training. We find that SSED performs over twice as well as the next best method in this setting.

To better understand the importance of using a VAE as a generative model we introduce SSED-EL, a version of SSED replacing the VAE with ACCEL's level editing strategy. SSED-EL may be viewed as an SSED variant of ACCEL augmenting $X_{\text{train}}$ using a non-parametric generative method, or equivalently as an ablation of SSED that does not approximate $p(\mathbf{x})$ and is therefore less grounded to the target CMDP. In Figure 4, we compare the two methods across level sets, and find that SSED improves more significantly over its ablation for level sets that are most similar to the original training set. This highlights the significance of being able to approximate $p(\mathbf{x})$ through the VAE to avoid distributional shift, which we discuss next.

## 6.2 DISTRIBUTIONAL SHIFT AND THE OVER-GENERALISATION GAP

Despite poor test scores, the UED baselines achieve small GenGap (as shown in Figure 8), as they perform poorly on both the test set and on $X_{\text{train}}$. Yet the fact they tend to perform well on the subset of $\mathbb{X}$ spanning their own training distribution means that they have over-generalised to an out-of-context set. As such, we cannot qualify their poor performance on $X_C$ as a lack of capability but instead as a form of misgeneralisation not quantifiable by the GenGap, and which are reminiscent of goal misgeneralisation failure modes reported in Di Langosco et al. (2022); Shah et al. (2022). Instead, we propose the *over-generalisation gap* as a complementary metric, which we define as

$$\text{OverGap}(\pi) \coloneqq \sum_{\tilde{\mathbf{x}} \in \Lambda} P_\Lambda(i_{\tilde{\mathbf{x}}}) \cdot V^\pi_{i_{\tilde{\mathbf{x}}}} - \frac{1}{|X_{\text{train}}|} \sum_{\mathbf{x} \in X_{\text{train}}} V^\pi_{i_{\mathbf{x}}}. \tag{10}$$

Note that OverGap compares the agent's performance with $X_{\text{train}}$ and as such is designed to measure over-generalisation induced by distributional shift.[2] Based on further analysis conducted in Appendix B.3, high OverGap coincides with the inclusion of out-of-context levels coupled with a significant shift in the level parameter distribution with respect to $p(\mathbf{x})$, and we find that SSED is the only level generation method tested able to maintain both low distributional shift and OverGap.

___
[2]using $X_{\text{test}}$ would make OverGap $\equiv$ GenGap if $P_\Lambda = \mathcal{U}(X_{\text{train}})$, whereas it should be 0.

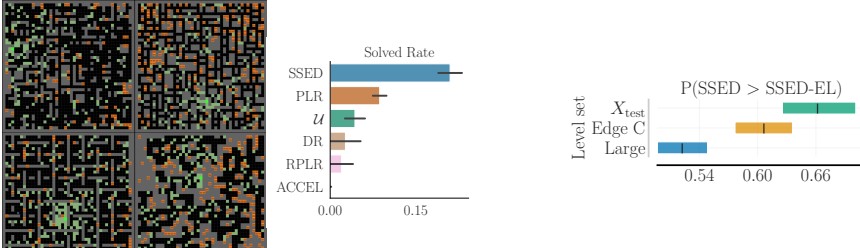

Figure 4: Left: zero-shot solved rate on a set of 100 levels with larger layouts. SSED's success rate is over twice PLR's, the next best performing method. Right: probability ($p < 0.05$) of SSED achieving higher zero-shot returns than its ablation SSED-EL, evaluated in each level set.

## 7 CONCLUSION

In this work, we investigated the impact of the level sampling process on the ZSG capabilities of RL agents. We found adaptive sampling strategies are best understood as data regularisation techniques minimising the mutual information between the agent's internal representation and the identity of training levels. In doing so, these methods minimise an upper bound on the generalisation gap, and our experiments showed it to act as an effective proxy for reducing this gap in practice. This theoretical framing allowed us to understand the mechanisms behind the improved generalisation achieved by value loss prioritised level sampling, which had only been justified empirically in prior work. We then investigated the setting in which the set of training levels is not fixed and where the generalisation bound can be minimised by training over an augmented set. We proposed SSED, a level generation and sampling framework restricting level generation to an approximation of the underlying distribution of a starting set of level parameters. We showed that this restriction lets SSED mitigates the distributional shift induced by UED methods. By jointly minimising the generalisation and over-generalisation gaps, we demonstrated that SSED achieves strong generalisation performance on in-distribution test levels, while also being robust to in-context edge-cases.

In future work, we plan to investigate how SSED scales to more complex environments. In a practical setting, the level parameter space is often high dimensional, and levels are described by highly structured data corresponding to specific regions of the parameter space. Depending on the simulator used, level parameters may consist of sets of values, configuration files or any other modality specific to the simulator. For example, they could be 3D scans of indoor environments (Li et al., 2021) or a vector map describing a city's road infrastructure (Wilson et al., 2021), which are often costly to collect or prescribe manually, and thus are limited in supply. Augmenting the number of training environments is therefore likely to play a role in scaling up RL in a cost effective manner. Our experiments show that unsupervised environment generation is problematic even in gridworlds, whereas the SSED framework is designed to scale with the amount of data being provided.

Lastly, we are interested in further exploring the synergies between SSED and mutual information minimisation frameworks. SSED performs data augmentation uptream of level sampling, whereas Jiang et al. (2021b) report significant improvements in combining PLR with data augmentation on the agent's observations (Raileanu et al., 2021) and thus acting downstream of level sampling. There may be further synergies in combining mutual information minimisation techniques at different points of the level-to-agent information chain, which is an investigation we leave for future work.

## 8 REPRODUCIBILITY STATEMENT

We believe parametrisable simulators are better suited to benchmark ZSG than procedural environments, as they provide a fine degree of control over the environment and are more consistent with a realistic application setting, as argued by Kirk et al. (2023). For example, our study of over-generalisation would not have been possible in Procgen, due to each game supporting a singular and non-parametrisable level distribution. Having access to the level parameters used in experiments also facilitates the reproducibility of ZSG research, and we make the train and evaluation sets of level parameters used in this work, as well as the code for running experiments, publicly available.[3] To encourage this practice in future work, we open-source our code[3] for specifying arbitrary CMDPs

---

[3]Available upon de-anonymisation of this publication

in Minigrid and generate their associated level sets, describing the generation process in detail in Appendix C. We also provide a dataset of 1.5M procedurally generated minigrid base layouts to facilitate level set generation.[3]

Proofs and derivations are included in Appendix A and additional implementation details, including the hyperparameters used and searched over are included in Appendix D.

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
