$$\approx \mathcal{H}(p(\mathbf{i})) + \frac{1}{|B|} \sum_{(i, H_t^o) \in B} \log p(\mathbf{i}|b(H_t^o)) \tag{12}$$

*where $\mathcal{H}(p)$ is the entropy of distribution $p$ and $B$ is a batch of trajectories collected from individual levels $i \in L$.*

*proof:*

*Given that the information chain of our model follows $H_t^o \rightarrow b \rightarrow f$, we have $MI(L, f \circ b) \leq MI(L, b)$ following the data processing inequality. $MI(L, b)$ can then be manipulated as follows*

$$MI(L, b) = \sum_{i \in L} \int d\mathbf{h} p(\mathbf{h}, i) \log \frac{p(\mathbf{h}, i)}{p(\mathbf{h})p(\mathbf{i})} \tag{13}$$

$$= -\sum_{i \in L} \int d\mathbf{h} p(\mathbf{h}, \mathbf{i}) \log p(i) + \sum_{i \in L} \int d\mathbf{h} p(\mathbf{h}, i) \log p(\mathbf{i}|\mathbf{h}) \tag{14}$$

$$\approx \mathcal{H}(p(\mathbf{i})) + \frac{1}{|B|} \sum_{n}^{|B|} \log p(i^{(n)}|\boldsymbol{h}^{(n)}) \tag{15}$$

*where in eq. (15) we approximate $p(\boldsymbol{h}, i)$ as the empirical distribution*

$$\tilde{p}(\boldsymbol{h}, i) = \begin{cases} \frac{1}{|B|} & \text{if } (H_t^o, i) \in B, \text{ with } \boldsymbol{h} = b(H_t^o) \\ 0 & \text{otherwise.} \end{cases} \tag{16}$$

*Note that we can treat non-recurrent architectures as a particular case, setting $H_t^o = o_t$ without loss of generality.*

## B   ADDITIONAL EXPERIMENTAL RESULTS

### B.1   PROCGEN ADDITIONAL EXPERIMENTAL RESULTS

From Equation (5), we estimate $MI(L, b)$ modelling $p_{\boldsymbol{\theta}}$ as a linear classifier. We plot this estimate against the GenGap and the $\ell_1$ value loss for all methods tested and across Procgen games in Figure 5. As expected under our theoretical framework, we measure a positive correlation between $MI(L, b)$ and GenGap with Kendall rank correlation coefficient $\tau = 0.53$ ($p < 1\mathrm{e}{-34}$), and a negative correlation with the $\ell_1$ value loss with Kendall rank correlation coefficient $\tau = -0.28$ ($p < 1\mathrm{e}{-16}$).

In order to provide a more intuitive quantification of our mutual information estimates, we consider the classification accuracy of the linear classifier used to compute our estimate for $MI(L, b)$, as these two quantities are proportional with each other. Out of 200 training levels, the classifier correctly predicts the current level $49\%$ of the times under uniform sampling, $34\%$ under $(S = S^V)$ and $23\%$ under $S^{\mathrm{MI}}$. Adaptive sampling strategies are therefore able to reduce $(S = S^{\mathrm{MI}})$ across ProcGen games, and ranking different methods according to their level classification accuracy will also sort them according to their respective GenGap. To understand how likely it is for a given sampling strategy to improve over another, we report the probability of improvement in test scores and GenGap for different pairs of strategies in Figure 6.

Nevertheless, the mean classifier accuracy remains 68 times random guessing for $(S = S^V)$ and 46 times random guessing for $(S = S^{\mathrm{MI}})$. As the classifier makes a prediction using an internal

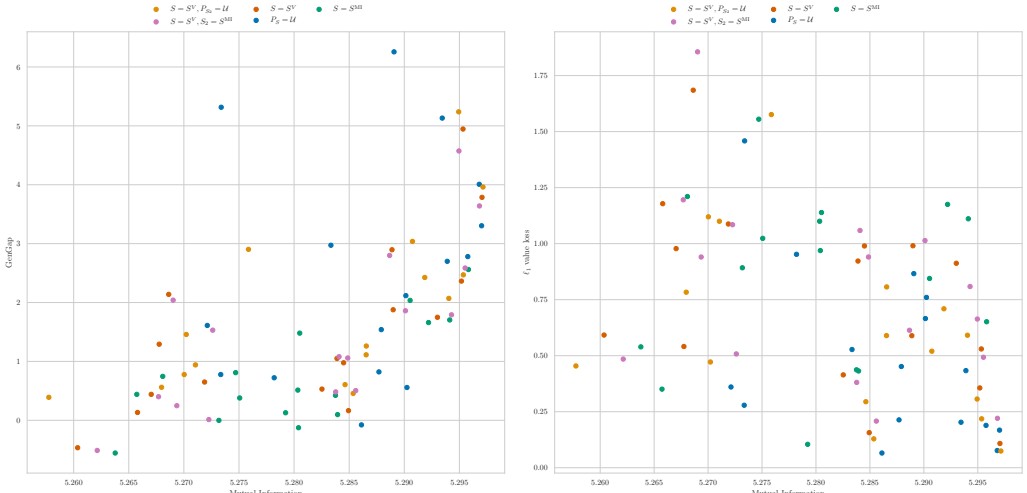

Figure 5: Scatter plot displaying the relationship between MI($L, b$) and the (unnormalised) GenGap (left) and with the $\ell_1$ average value loss (right), measured across all methods and Procgen games at the end of training. Each point represents 5 seeds of a level sampling method in a particular game.

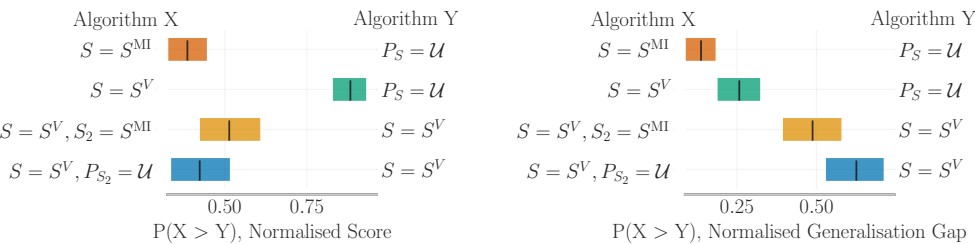

Figure 6: Probability of algorithm $X$ incurring a higher normalised test score (left) and GenGap (right) than algorithm $Y$. Evaluation performed over 5 seeds across all Procgen games, using the rliable library(Agarwal et al., 2021). Colored bars indicate the 95% confidence interval.

representation obtained from a single observation we find these results surprising, and demonstrate adaptive sampling strategies can only reduce MI($L, b$) up to a point. To further reduce MI($L, b$), adaptive sampling should be combined with other data regularisation techniques, such as the level augmentation technique proposed by SSED and/or additional data augmentation techniques. Indeed Jiang et al. (2021b) report a significant improvement in test scores when combining PLR with UCB-DrAC Raileanu et al. (2021), an observation augmentation method.

### B.2 COMPARING THE EFFECTIVENESS OF ADAPTIVE SAMPLING STRATEGIES ACROSS PROCGEN GAMES

We observe that both the classifier accuracy under uniform level sampling and the potential improvement induced by adaptive sampling is highly dependent on the procgen game tested. To better understand why, we compare the measured accuracy with a qualitative analysis of the observations and levels encountered in the Maze and Bigfish games, which we provide a sample of in Figure 7.

In Maze, the accuracy remains over $80\%$ ($160\times$ random) for all methods tested and the reduction is GenGap insignificant. On the other hand, in Bigfish all adaptive sampling strategies tested lead to a significant reduction in classifier accuracy, dropping from $80\%$ to under $20\%$, and they are associated with a significant drop in GenGap and improvement in test scores.

In Maze, the observation space is set up such that the agent observes the full layout at each timestep. The layout is unique to each level and provides many features for identification that are straightfor-

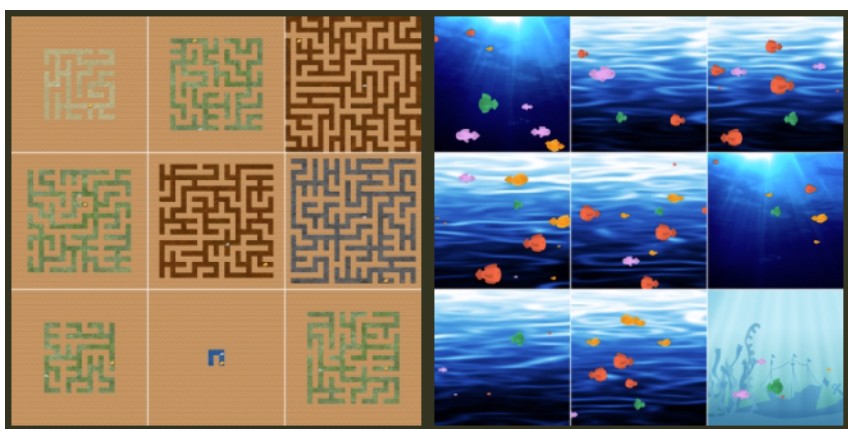

Figure 7: Agent observations sampled from 9 levels from the Maze (left) and Bigfish (right) games of the Procgen benchmark.

ward to learn by the agent's ResNet architecture. In addition, these features cannot be ignored by the agent model in order to solve the task. Intuitively, we can hypothesise that adaptive sampling strategies will not be effective if all the levels are easily identifiable by the agent, which appears to be the case in Maze. In these cases, other data regularisation techniques, such as augmenting the observations, can be more effective, and in fact Jiang et al. (2021b) report that Maze is one of the games where combining PLR with UCB-DrAC leads to a significant improvement in test scores.

On the other hand, we observe that many of the Bigfish levels yield similar observations. Indeed, both the features relevant to the task (the fish) and irrelevant (the background) are similar in many of the training levels. Furthermore, there's significant variation in the observations encountered during an episode, as fish constantly appear and leave the screen. Yet, some levels (top left, middle and bottom right) are easily identifiable thanks to their background, and we can hypothesise that adaptive sampling strategies will tend to de-prioritise them more often, essentially performing data regularisation via a form of rejection sampling.

### B.3 QUANTIFYING THE DISTRIBUTIONAL SHIFT IN MINIGRID

In Figure 8, we report the generalisation and over-generalisation gaps in the Minigrid experiments. We observe that UED methods tend to exhibit lower generalisation gaps than SSED, PLR or uniform sampling. We find that introducing an additional metric in the form of the OverGap Equation (10) necessary to quantify this form of misgeneralisation. We next study the correlation between the OverGap and distributional shifts between the underlying CMDP level parameter distribution $p(\mathbf{x})$ and $p_\Lambda(\mathbf{x})$, the distribution of level parameters existing in the level replay buffer $\Lambda$.

We approximate $p(\mathbf{x})$ as $\tilde{p}(\mathbf{x}) \approx \mathcal{U}(X_{\text{train}})$ and we use the Jensen-Shannon Divergence (JSD) between $\tilde{p}(\mathbf{x})$ and $p_\Lambda(\mathbf{x})$, defining the JSD in a space consistent with the CMDP semantics. To do so, we measure the distribution of distances between the goal location and each other tile type. The JSD is therefore expressed as $\text{JSD}(c_p||c_q)$, where $c(t, d|\mathbf{x})$ is the categorical distribution measuring the probability of tile type $t$ occurring at distance $d$ from the goal location in a given level $\boldsymbol{x}$ and $c_p$ is the marginal $c_p = \mathbb{E}_{\mathbf{x} \sim p(\mathbf{x})}[c(t, d|\mathbf{x})]$, setting $p_\Lambda(\mathbf{x})$ and $q = \mathcal{U}(X_{\text{train}})$.

We report how the JSD evolves over the course of training for different methods in Figure 8c. We observe that distributional shift occurs early on during training and remains relatively stable afterwards in all methods. JSD and OverGap tend to be positively correlated for most methods, except for DR and PLR, which both present high JSD but low OverGap. SSED is the only generative method to maintain a low JSD throughout trainingIn fig. 9, we report additional metrics on the levels sampled by each method. We find that SSED tends to be as proficient as PLR in maintaining consistency with $X_{\text{train}}$ with the occurence of different tile types, as well as higher order task-relevant properties such as the shortest path length between the start and goal locations.

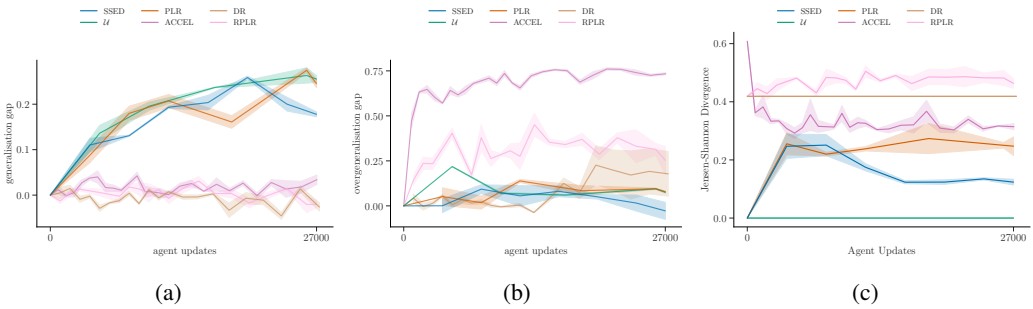

(a)                                    (b)                                    (c)

Figure 8: Generalisation gap ((a), Equation (1)) and over-generalisation gap ((b), Equation (10)) of different methods during training. Fixed set sampling strategies experience higher generalisation gap, while UED methods are dominated by the over-generalisation gap. SSED tends to follow a similar profile as fixed set sampling methods, with a moderate generalisation gap and a low (and even at times negative) over-generalisation gap. SSED also exhibit small distributional shift in its level parameters, as demonstrated by the evolution of the JSD over the course of training (c). Surprisingly, SSED demonstrates a smaller divergence than PLR, even when PLR only has access to $X_{\text{train}}$ and as such can only affect the JSD by changing the prioritisation of individual levels $i_{\boldsymbol{x}}$, $\boldsymbol{x} \in X_{\text{train}}$.

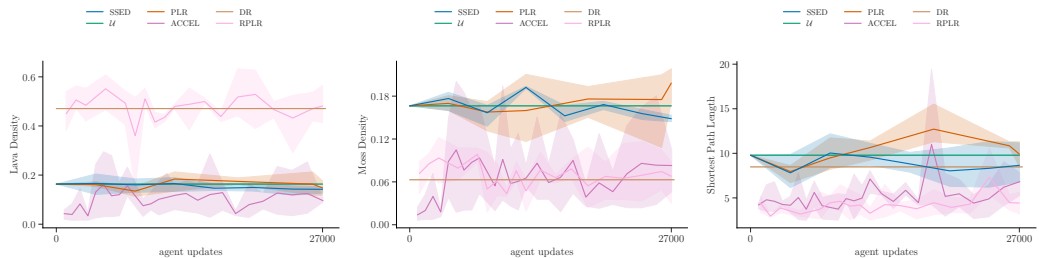

Figure 9: Left and middle: evolution of lava and moss tile densities encountered in sampled levels over the course of training. Right: evolution of the shortest path length between the start and goal location in sampled levels over the course of training.

## C  CMDP SPECIFICATION AND THE LEVEL GENERATION PROCESS

In Minigrid Chevalier-Boisvert et al. (2018), the agent receives as an observation a partial view of its surroundings (in our experiments it is set to two tiles to each side of the agent and four tiles in front) and a one-hot vector representing the agent's heading. The action space consists of 7 discrete actions, however in our setting only the actions moving the agent forward and rotating it to the left or right have an effect on the environment. The episode starts with the agent at the start tile and facing a random direction. The episode terminates successfully when the agent reaches the goal tile and receives a reward between 0 and 1 based on the number of timesteps it took to get there. The episode will terminate without a reward if the agent steps on a lava tile, or when the maximum number of timesteps is reached.

Levels are parameterised as 2D grids representing the overall layout, with each tile type represented by an unique ID. Tiles can be classified as navigable (for example, moss or empty tiles) or non-navigable (for example, walls and lava, as stepping into lava terminates the episode). To be valid, a level must possess exactly one goal and start tile, and to be solvable there must exist a navigable path between the start and the goal location. We provide the color palette of tiles used in Figure 12.

In this work, we define and train within the "Cave Escape" CMDP, which corresponds to a subset of the solvable levels in which moss and lava node placement is respectively positively and negatively correlated with the geodesic distance to the goal.[4] Under partial observability, the minimax regret

___

[4]To measure the distance-to-goal of a non-navigable node, we first find the navigable node that is closest from it and measure its geodesic distance to the goal. We then add to it the distance between this navigable

policy for this CMDP would leverage moss and lava locations as context cues, seeking regions with perceived higher moss density avoiding regions with perceived high lava density, and thus the CMDP possesses an attributable goal and optimal behavior. We provide example levels of the CMDP in Figure 10.

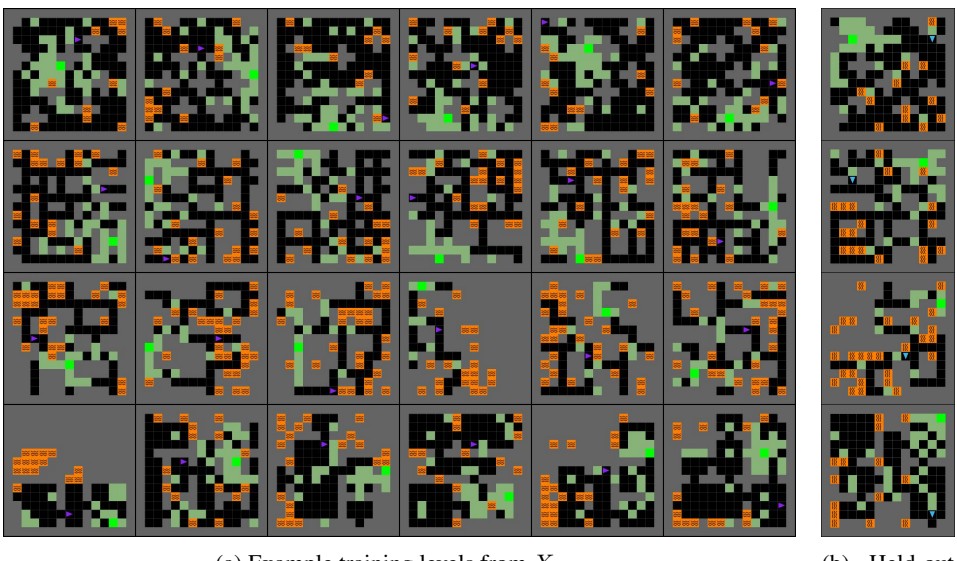

(a) Example training levels from $X_{\text{train}}$

(b) Held-out levels

Figure 10: Sample levels from $X_{\text{train}}$ and from held-out test levels. Wall tiles are rendered in gray, empty tiles in black, moss tiles in green and the goal tile in lime green. The agent is rendered as a blue or purple triangle, and is depicted at its start location. Each row corresponds to levels generated with a specific wave function collapse base pattern. Four different base patterns are used in $X_{\text{train}}$.

## C.1 GENERATING HIGHLY STRUCTURED LEVELS

We employ the wave function collapse (WFC) algorithm (Gumin, 2016) as our procedural generation algorithm to obtain highly structured but still diverse gridworld layouts. WFC takes as an input a basic pattern and gradually collapses a superposition of all possible level parameters into a layout respecting the constraints defined by the input pattern. By doing so, it is possible to generate a vast number of tasks from a small number of starting patterns. Given suitable base patterns the obtained layouts provide a high degree of structure, and guarantee that both task structure and diversity scale with the gridworld dimensions. We provide 22 different base patterns and allow for custom ones to be defined. After generating a layout using WFC, we convert the navigable nodes of a layout into a graph, choose its largest connected component as the layout and convert any unreachable nodes to non-navigable nodes. We place the goal location at random and place the start at a node located at the median geodesic distance from the goal in the navigation graph. By doing so we ensure that the complexity of generated layouts is relatively consistent given a specific grid size and base pattern. Finally we sample tiles according to parameterisable distributions defined over the navigable or non-navigable node sets. In this work the tile set consists of the { moss, empty, start, goal } tiles as the navigable set and the { wall, lava} tiles as the non-navigable set, and we define distributions for the moss and lava node types over the navigable and non-navigable node sets, respectively.

## C.2 CONTROLLING LEVEL COMPLEXITY

We provide two options to vary the complexity of the level distribution. The first is to change the gridworld size, which directly results in an increase in complexity. The second, which is specific to the Cave Escape CMDP, is to change the sampling probability of moss and lava nodes. Since the

---

node and the non-navigable node of interest. If there are multiple equally close navigable nodes, we select the navigable node with the smallest geodesic distance to the goal.

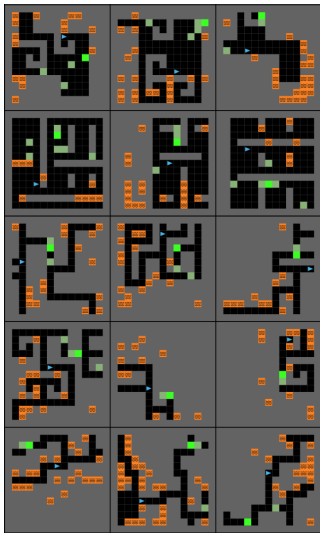 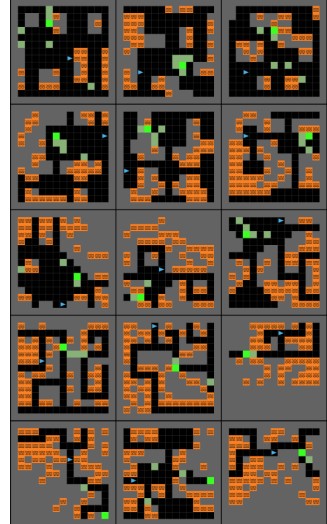

(a) Example levels from the first set of edge cases

(b) Example levels from the second set of edge cases

Figure 11: We generate 2 separate sets of edge cases, using 14 different base patterns not used to generate $X_{\text{train}}$. The moss density of the first set (a) is 3 times as small as in $X_{\text{train}}$, making finding the goal using CMDP contextual cues more challenging. (b) is the same as (a) but also has the lava density multiplied by a factor of 3, making it more difficult to avoid walking into lava and failing the episode. Both sets are combined when evaluating the agent on edge cases.

environment is partially observable, reducing the fraction of moss to navigable nodes, or increasing their entropy diminishes the usefulness of moss tiles as context cues. On the other hand, increasing the density of lava tiles increases the risk associated with selecting the wrong action during play. Thus it is straightforward assess the agent's performance on edge-cases by defining level sets with a larger layout size, or with shifted moss and lava tile distributions.

## D  IMPLEMENTATION DETAILS

### D.1  PROCGEN

The Procgen Benchmark is a set of 16 diverse PCG environments that echo the gameplay variety seen in the ALE benchmark Bellemare et al. (2015). The game levels, determined by a random seed, can differ in visual design, navigational structure, and the starting locations of entities. All Procgen environments use a common discrete 15-dimensional action space and generate $64 \times 64 \times 3$ RGB observations. A detailed explanation of each of the 16 environments is given by Cobbe et al. (2020). Leading RL algorithms such as PPO reveal significant differences between test and training performance in all games, making Procgen a valuable tool for evaluating generalisation performance.

We conduct our experiment on the easy setting of Procgen, which employs 200 training levels and a budget of 25M training steps, and evaluate the agent's ZSG performance on the full range of levels, excluding the training levels. We calculate normalised test returns using the formula $\frac{(R - R_{\min})}{(R_{\max} - R_{\min})}$, where $R$ is the non normalised return and $R_{\min}$ and $R_{\max}$ are the minimum and maximum returns for each game as provided in (Cobbe et al., 2020).

We employ the same ResNet policy architecture and PPO hyperparameters used across all games as Cobbe et al. (2020), which we reference in Table 1. To compute the MI based scoring strategy $S^{\text{MI}}$ used in our experiments, we parametrise $p_{\boldsymbol{\theta}}(i|b(o_t))$ as a linear classifier and we ensure the training processes of the agent and the classifier remain independent from one-another by employing a separate optimiser and stopping the gradients from propagating through the agent's network.

### D.2 MINIGRID RL AGENT

We use the same PPO-based agent as reported in Parker-Holder et al. (2022). The actor and critic share the same initial layers. The first initial layer consists of a convolutional layer with 16 output channels and kernel size 3 processes the agent's view and a fully connected layer that processes its directional information. Their output is concatenated and fed to an LSTM layer with hidden size 256. The actor and critic heads each consist of two fully connected layers of size 32, the actor outputs a categorical distribution over action probabilities while the critic outputs a scalar. Weights are optimized using Adam and we employ the same hyperparameters in all experiments, reported in Table 1. Trajectories are collected via 36 worker threads, with each experiment conducted using a single GPU and 10 CPUs.

Following Parker-Holder et al. (2022), the non dataset based methods employ domain randomisation as their standard level generation process, in which the start and goal locations, alongside a random number between 0 and 60 moss, wall or lava tiles are randomly placed. The level editing process of ACCEL and SSED-EL remains unchanged from Parker-Holder et al. (2022), consisting of five steps. The first three steps may change a randomly selected tile to any of its counterparts, whereas the last two are reserved to replacing the start and goal locations if they had been removed in prior steps.

We train three different seeds for each baseline. We use the same hyperparameters as reported in Parker-Holder et al. (2022) for the DR, RPLR and ACCEL methods and as Jiang et al. (2021b) for PLR, as an extensive hyperparameter search was conducted in a similar-sized Minigrid environment in each case. SSED employs the same hyperparameters as PLR for its level buffer, with the additional secondary sampling strategy hyperparameters introduced by SSED. We did not perform an hyperparameter search for these additional hyperparameters as we found that the initial values worked adequately. We report all hyperparameters in Table 1.

### D.3 VAE ARCHITECTURE AND PRE-TRAINING PROCESS

We employ the $\beta-$VAE formulation proposed in Higgins et al. (2017), and we parametrise the encoder as a Graph Convolutional Network (GCN), a generalisation of the Convolutional Neural Network (CNN) (Krizhevsky et al., 2012) to non Euclidian spaces. Our choice of a GCN architecture is motivated by the fact that the level parameter space $\mathbb{X}$ is simulator-specific. Employing a graph as an input modality for our encoder gives our model additional flexibility to by applicable to different simulators. Using a GCN, some of the inductive biases that would be internal in a traditional architecture can be defined through the wrapper encoding the environment parameter $x$ into the graph $\mathcal{G}_x$. For example, in minigrid, we represent each layout as a grid graph, each gridworld cell being an individual node. Doing so makes our GCN equivalent to a traditional CNN in the Minigrid domain. We select the GIN architecture (Xu et al., 2019) for the GCN, which we connect to an MLP network that outputs latent distribution parameters $\mu_z, \sigma_v z$. The decoder is a fully connected network with three heads. The *layout* head outputs the parameters of Categorical distributions for each grid cell, predicting the tile identity between [Empty, Moss, Lava, Wall]. The *start* and *goal* heads output the parameters of Categorical distributions predicting the identity of the start and goal locations across grid cells, which matches the inductive bias of a single goal or start node being present in any given level.

We pre-train the VAE for 200 epochs on $X_{\text{train}}$, using cross-validation for hyperparameter tuning. During training, we formulate the reconstruction loss as a weighted sum of the cross-entropy loss for each head.[5] At deployment, we guarantee *valid* layouts (layouts containing a unique start and goal location, but not necessarily solvable) by masking the non-passable nodes sampled by the layout head when sampling the start location, and masking the generated start and non-passable nodes when sampling the goal location. In this way, we guarantee unique start and goal locations that will not override one-another. Note that our generative model may still generate *unsolvable* layouts, which do not have a passable path between start and goal locations, and therefore it must learn to generate solvable layouts from $X_{\text{train}}$ in order to be useful. We do not explicitly encourage the VAE to generate solvable layouts, but we find that optimising for the ELBO in Equation (9) is an effective proxy. Layouts reconstructed from $X_{\text{train}}$ have over 80% solvability rate, while layouts generated

---

[5]To compute the cross-entropy loss of the layout head, we replace the start and goal nodes in the reconstruction targets by a uniform distribution across {moss, empty}.

via latent space interpolations have over 70% solvability rate. This indicate that maximising the ELBO results in the generative model learning to reproduce the high-level abstract properties that are shared across the level parameters in the training set, which we also observe in Figure 9.

We conduct a random sweep over a budget of 100 runs, jointly sweeping architectural parameters (number of layers, layer sizes) and the $\beta$ coefficient, individual decoder head reconstruction coefficients and the learning rate. Each run takes 20 minutes, which means that our sweeping procedure takes less time to complete than a single seed of our Minigrid experiment. We report the chosen hyperparameters in Table 2.

Table 1: Hyperparameters used for Minigrid experiments. Hyperparameters shared between methods are only reported if they change from the method above.

| Parameter | Procgen | MiniGrid |
|---|---|---|
| *PPO* | | |
| $\gamma$ | 0.999 | 0.995 |
| $\lambda_{\text{GAE}}$ | 0.95 | 0.95 |
| PPO rollout length | 256 | 256 |
| PPO epochs | 3 | 5 |
| PPO minibatches per epoch | 8 | 1 |
| PPO clip range | 0.2 | 0.2 |
| PPO number of workers | 64 | 32 |
| Adam learning rate | 5e-6 | 1e-4 |
| Adam $\epsilon$ | 1e-5 | 1e-5 |
| PPO max gradient norm | 0.5 | 0.5 |
| PPO value clipping | yes | yes |
| return normalisation | yes | no |
| value loss coefficient | 0.5 | 0.5 |
| student entropy coefficient | | 0.0 |
| generator entropy coefficient | | 0.0 |
| | | |
| *PLR* | | |
| Scoring function | | $\ell_1$ value loss |
| Replay rate, $p$ | | 1.0 |
| Buffer size, $K$ | | 512 |
| Prioritisation, | | rank |
| Temperature, | | 0.1 |
| Staleness coefficient, $\rho$ | | 0.3 |
| | | |
| *RPLR* | | |
| Scoring function, | | positive value loss |
| Replay rate, $p$ | | 0.5 |
| Buffer size, $K$ | | 4000 |
| | | |
| *ACCEL* | | |
| Edit rate, $q$ | | 1.0 |
| Replay rate, $p$ | | 0.8 |
| Buffer size, $K$ | | 4000 |
| Edit method, | | random |
| Levels edited, | | easy |
| | | |
| *SSED* | | |
| Replay rate, $p$ | | 1.0 |
| Scoring function support, | | dataset |
| Staleness support, | | dataset |
| Secondary Scoring function, | | $\ell_1$ value loss |
| Secondary Scoring function support, | | buffer |
| Secondary Temperature, | | 1.0 |
| Mixing coefficient, $\eta$ | | linearly increased from 0 to 1 |

Table 2: Hyperparameters used for pre-training the VAE.

| Parameter | |
| --- | --- |
| *VAE* | |
| $\beta$ | 0.0448 |
| layout head reconstruction coefficient | 0.04 |
| start and goal heads reconstruction coefficients | 0.013 |
| number of variational samples | 1 |
| Adam learning rate | 4e-4 |
| Latent space dimension | 1024 |
| number of encoder GCN layers | 4 |
| encoder GCN layer dimension | 12 |
| number of encoder MLP layers (including bottleneck layer) | 2 |
| encoder MLP layers dimension | 2048 |
| encoder bottleneck layer dimension | 256 |
| number of decoder layers | 3 |
| decoder layers dimension | 256 |



Figure 12: Color palette used for rendering minigrid layouts in this paper and their equivalent for Protanopia (Prot.), Deuteranopia (Deut.) and Tritanopia (Trit.) color blindness. We refer to each row in the main text as, in order: green (goal tiles), pale green (moss tiles), blue (agent), black (empty/floor tiles), grey (wall tiles) and orange (lava tiles).