# OpenReview forum: "How the Level Sampling Process impacts Zero-Shot Generalisation in Deep Reinforcement Learning"
_ICLR.cc/2024/Conference — Submitted to ICLR 2024_

### Official Review · Reviewer_zMV7 · 2023-10-19

**Soundness:** 3 good
**Presentation:** 2 fair
**Contribution:** 3 good
**Rating:** 6
**Confidence:** 3

**Summary:**

This paper considers auto-curriculum learning over the minigrid testbed and aims to train a visual policy that can generalize across various generated minigrid levels in a zero-shot manner. The paper first demonstrates the insights that the mutual information between representations and the training level. Then the paper further develops an auto-curriculum algorithm that leverages a smart sampling strategy using a pretrained VAE sampler. By combining both parts, the experiment results suggest strong zero-shot visual generalization can be achieved.

**Strengths:**

1. I really **appreciate the discussions in Section 4** on the connection of generalization capability and the mutual information between the representation and level identity, which is insightful. The discussion makes the proposed MI criterion natural and intuitive. Although the paper can be much stronger by presenting proof rather than simply stating it as a hypothesis.

2. **The experiment results look strong**. I also appreciate the discussion on the importance of VAE, which should be crucial intuitively considering the testbed is minigrid. Btw, I personally guess that VAE might be unnecessary if you adopt another testbed with a goal-conditioned flavor (e.g., those environments where you create a new instance by setting a new goal).

**Weaknesses:**

1. **The presentation can be improved**. There are many notations that are introduced without definitions. For example, in Theorem 3.1 you didn't introduce $L$ (although I can understand its meaning after reading the whole paper). In the paragraph after equation (3), $\hat{V}_t$ is not defined either. The authors seem to have a strong intention to pack a lot of knowledge in Section 3, ranging from notation to existing theorems and algorithms. Section 3 looks a bit messy to me and hard to follow if the reader is not an expert who _masters_ all the related works. I think the section can be much better organized and self-contained. You may want to have some sub-sections with some high-level mathematical descriptions of previous works (in addition to the related work section).

2. **citation issues:** Most related works cited in this paper are within the past 3 years. I think the authors ignore a large portion of works in curriculum learning literature, such as those working on goal generation, open-ended learning, and multi-task learning. Although these works do not work on the visual minigrid test, they do share a similar high-level principle to this work and should be acknowledged. For example, the paper states "_we find that strategies de-prioritising levels with low value loss, as **first** proposed in prioritised level replay_". Well, I have to say in curriculum learning, many works have leveraged the idea of using value function as an indicator for prioritization. [Here](https://proceedings.neurips.cc/paper/2020/file/566f0ea4f6c2e947f36795c8f58ba901-Paper.pdf) is an example. I think you can do a brief survey to get more.

2. **minor issues:** Fig 2 is derived from the results of a non-recurrent policy, as stated in the paragraph below equation (7). Why not use an LSTM, as what you have done in the experiments?

**Questions:**

Although I personally keep a positive perspective on this paper, I would still expect the authors to update the paper for an improved presentation, which can make my judgment firm.

It would be also great if the authors could further provide more analysis or even theoretical analysis of the hypothesis.

---

> ### Author Response · Authors · 2023-11-22
>
> We thank the reviewer for his feedback and positive comments on our paper. We hope they will appreciate even more the new and improved section 4! In particular, we point to sections 4.1 and 4.2 which provides a more rigorous analysis of our hypothesis.
>
> We believe the reviewer's points have been addressed in our post titled titled "Summary of changes in updated version" which gives a comprehensive overview of the changes made in the new revision. We have fixed the notation issues pointed out by the reviewer, and we hope the structure added to section 3 makes it now easier to follow.

---

### Official Review · Reviewer_mvxM · 2023-10-31

**Soundness:** 2 fair
**Presentation:** 2 fair
**Contribution:** 3 good
**Rating:** 3
**Confidence:** 3

**Summary:**

The paper aims to connect a working technique in active domain randomization with a bound on mutual information between histories and contexts/levels. The authors also introduce a method (SSED) to sample more contexts from a similar distribution than the training set by training a VAE on the context-parameters. Both claims are evaluated on maze navigation tasks, which have been modified such that the optimal policy on the training set is not the same as on the entire parameter set. Results show that SSED improves the performance while reducing the optimality gap slightly (but significantly).

**Strengths:**

The paper introduces a, to the best knowledge of this reviewer, novel concept of over-generalization to the space of all possible parameters. The proposed method SSED is not terribly novel, but makes sense and shows the effect nicely. The open sourced maze environment is another good contribution of the paper.

**Weaknesses:**

**TLDR:** the paper is interesting, but in a bad state. On the one hand, it is very confusingly written, overpromises and does not do all it claims. The first contribution is also dubious: the derivation is incomplete (many terms just appear without definition), seem to rely too much on intuition, and the results do not show the effect. On the other hand, the second contribution (SSED) is simple but interesting, and the results demonstrate that it works. Nonetheless, the reviewer cannot recommend to accept the paper in its current form.

1. The paper overpromises many things, for example from the abstract:
	- "As a first step, we measure the mutual information (MI) between the agent’s internal representation and the set of training levels, which we find to be well-correlated to instance overfitting": there is no empirical measurement of MI or its correlation with overfitting in the paper. All evidence is circumstantial and the interpretation of Figure 2, in this reviewer's opinion, wrong.
	- "adaptive sampling strategies prioritising levels based on their value loss are more effective at maintaining lower MI", "We then turn our attention to unsupervised environment design (UED) methods, which [..] minimise MI more effectively than methods sampling from a fixed set.", "SSED generates levels using a variational autoencoder, effectively reducing MI": again, MI is never measured and the claim that the evaluated methods minimize MI is pure conjecture.

2. The paper is confusingly written and introduces many terms without defining them. For example, $\text{MI}(i,\pi)$ is never defined in the main paper. $S^V_i$ is defined with $\hat V_t$ and $V_t$, which have never been defined. On Page 5 the authors state $V_i^\pi(s) = V^\pi(s) + v_i^\pi(s)$, but $V^\pi(s)$ has never been defined (only for a general POMDP, the CMDP only knows context-specific values called $V_{i_x}^\pi$). Sometimes the used PPO baseline uses non-recurrent intermediate representations (p.5), another time the same implementation uses an LSTM-based architecture (which is recurrent, p.7). The reviewer has the impression that many of these details could be understand upon reading the cited papers, but as is, the paper does a poor job explaining them.

3. The connection between generalization, MI and scoring the value loss ($S^V$) is indirect, and not rigorous enough. The connection between MI and generalization is not very clear: in Figure 1 both following the black and the green cells leads to the goal in (a) and (b), but only one of the two generalizes to (c). Both have thus the same MI, but different generalization. While the statement "An agent learning level-specific policies implies high MI between its internal representation and the level identities" (p.1-2) makes generally sense, the conclusion that agents with high MI "will not transfer zero-shot to new levels" is less clear. Moreover, it is not apparent why the negation *agents with low MI generalize well*, which seem to be the basis for this paper, should be true. So the only clear connection seems to be the bound in (eq.2). The reviewer also missed a connection why "Sampling levels associated with highly negative classifier cross-entropies therefore results in less mutual information" (p.5). Finally, the idea that $V_i^\pi(s) = V^\pi(s) + v_i^\pi(s)$ separates into two errors and the second is somehow connected to MI (without any clarification how exactly) is flawed, as the errors of $V^\pi$ and $v_i^\pi$ could also cancel each other out, yielding low error, but high MI.

4. The connection of MI to generalization and to the value-error scoring is not supported by the presented data in Figure 2. Here the first two rows are not significantly different, which would lead to the conclusion that using $S_2=S^{MI}$ does *not* change the algorithm's behavior much. How does this justify the statement "$S = S^V$ therefore appears to strike a good balance between sample efficiency and mutual information minimisation", if MI is not necessary for performance. The third row shows $S^{MI}$ having a smaller generalization gap, but the performance is also much smaller, and these two metrics are highly correlated! The results can therefore also be interpreted as "using MI reduces the performance, and *as a consequence* has a smaller generalization gap". For the same reason the second plot to the right is useless without knowing the algorithms' performances.

5. The entire Section 4 should be either removed or replaced by an experiment that actually links generalization (not just the generalization gap) to MI, demonstrates or proves that sampling levels with high MI *reduces* the overall MI, and shows a correlation between MI and the value-error score.

**Questions:**

- The approximation in Lemma 4.1 only works if the levels are constantly sampled with $p(i)$, but your method aims to change this distribution. How does this work when your scoring selects the levels that make up the batch $B$?
- Your goal is to produce policies that have low MI on the training set, but approximated MI is not used to change the policy, only the sampling of levels. Is there a reason why you do not simply reward actions with lower MI?

---

> ### Author Response · Authors · 2023-11-22
>
> We thank the reviewer for thoroughly reviewing our paper and offering detailed feedback for its improvement. We believe the new revision addresses the weaknesses that were pointed out. We invite the reviewer to refer to our post titled "Summary of changes in updated version" for a comprehensive overview of the changes made, which we connect to the reviewer's feedback below.
>
> Addressing weaknesses:
>
> 1. `there is no empirical measurement of MI or its correlation with overfitting in the paper.` and `MI is never measured`.
> In section 4.3 we include an additional result measuring the correlation of our estimate for $MI(L,b)$ with the generalisation gap and with the value loss, with additional analysis conducted in appendix B.1
>
> 2. We thank the reviewer for pointing out these missing definitions, which are now properly introduced in the new revision. Regarding recurrent and non-recurrent representations, we now clarify how our analysis is equally applicable to recurrent and non-recurrent architectures in section 4.1.
>
> 3.
> a) We hope our restructuring of section 4 helps in understanding the connection between $S^V$, $MI(L,b)$ and the generalisation gap. We agree with the reviewer that agents with high MI will not always generalise poorly, as MI characterises an *upper bound* on the generalisation gap, while the gap itself may be small if that bound is not tight. However, our experiments results reported in section 4.3 and appendix B.1 and B.2 indicate that, in practice, high MI does imply a significant generalisation gap.
>
> b) To clarify, we are not making the claim that "agents with low MI generalise well". We claim that "agents with low MI have a small generalisation gap". This does not always imply strong generalisation. For example, achieving train and test scores of zero yield a zero generalisation gap, however the agent cannot be described as generalising well. To an extent, this example is an extreme version of what happens under the $S=S^{MI}$ sampling strategy, where the generalisation gap is reduced by adversely impacting the performance on the train set. We hope the updated section 4, and in particular section 4.3 and the updated Figure 2 are now clearer in explaining this failure mode induced by certain mutual information minimisation strategies.
>
> c) Section 4.2 now focuses exclusively on motivating the $V_i^\pi(s) = V^\pi(s) + v_i^\pi(s)$ decomposition and its relationship to MI. the unbiased value estimator lemma from Bertran et al. (2020), included as Lemma 3.1 in the new revision, states that $\mathbb{E}_L[V^\pi_i]$ is an *unbiased* estimator for the CMDP value function (redefined as $V^\pi_C$ in the new version for added clarity) under any set of levels $L$. As $\hat{V}$ is trained to predict $V^\pi_i$ under $L$, at convergence it is an unbiased estimate of $V^\pi_C$. Finally, the bias-variance trade-off tells us $\hat{V}$ will tend to have low bias but high variance as it starts overfitting. In fact, we may informally view $v^\pi_i$ as a level-specific "noise function". In order to accurately predict this "noise function" $v^\pi_i$, $\hat{V}$ needs to be learn to identify level identities, as different levels have different $v^\pi_i$. Hence $MI(L,b)$ becoming high as the value loss goes to zero.
>
> 4. and 5. we believe to have addressed these points in the comments above and invite the reviewer to also refer to the updated section 4 and to the new Figure 2.
>
> Answering questions:
>
> 1. Prioritised sampling effects the model by changing the training data. However the resulting trained model must be evaluated over the context distribution associated to the CMDP and MI also must be measured over this distribution. In procgen, all train seeds correspond to in-context levels and can be considered equally likely in the CMDP contextual distribution. Therefore the empirical estimation for that distribution is the distribution giving all training levels equal weights. We emphasise this distinction in section 4.1 of the new revision.
>
> 2. In this work, we chose to focus specifically on the level sampling process as we view it as a form of data regularisation that is understudied when compared to the number of existing contributions tackling ZSG through modifications of the reward function or the agent architecture, or by augmenting the observations. However, as our method is orthogonal to such approaches there may be interesting synergies to explore in future work.

---

### Official Review · Reviewer_GmmJ · 2023-10-31

**Soundness:** 3 good
**Presentation:** 4 excellent
**Contribution:** 3 good
**Rating:** 8
**Confidence:** 4

**Summary:**

One of the fundamental problems in reinforcement learning is generalization of the learned policies to new environments. One solution approach is to use an adaptive sampling strategy over a wide range of environments. However, the level of sampling required to achieve the desired generalization remains unknown. The authors propose a new theoretical framework to answer this question using mutual information and minimization of an upper bound on the generalization error from adaptive sampling. Once this relation is established, now, the authors study the problem of creating new environments systematically and improve generalization. Specifically, the authors propose a self-supervised environment design (SSED) to minimize mutual information for zero-shot generalization. The authors provide a theoretical bound and proof for the generalization gap using mutual information with training level and reinforcement learning policy. Then, use level scores from rollout trajectories to define adaptive sampling distribution.

SSED consists of two components: a generative phase, in which a variational autoencoder (VAE) is employed as a generative model and a replay phase, in which we use an adaptive distribution to sample levels. The algorithm alternates between the generative and replay phases, and only perform gradient updates on the agent during the replay phase, while the VAE weights remain fixed throughout training. The authors use a complex environment benchmark, ProcGen from OpenAI, and Minigrid, a gridworld navigation domain (ChevalierBoisvert et al., 2018) for empirical evaluation.

**Strengths:**

Originality & Significance. In some sense, it feels like a no-brainer to use variability in the new environment setup and maximize diversity (minimize mutual information) for better generalization. Similar to classical system identification methods for control systems. I believe the originality comes off from measuring the diversity in the environment to quantify generalization, instead of randomly exploring over a large set of environments.

Quality & Clarity. The paper is well-written. And the explanations are clear. There are not many grammatical errors.

The authors baseline their approach to other state-of-the-art approaches.

**Weaknesses:**

The authors originally use ProcGen to describe some of the concepts but later on all the empirical experiments are in Minigid. While Minigrid is a good toy problem to start with, it lacks the complexity of the most real-world environments where the generalization is the most important. Having to include ProcGen examples would have been a good mid-step towards addressing real-world challenges in generalization to new environments.

**Questions:**

I was surprised that while the authors started out with ProcGen, then switched to Minigrid. While the Minigrid provides a good framework as a starting point to showcase the generalization issue, I believe it lacks many of the real-world complexities for generalizing the RL policies to new environments.

---

> ### Author Response · Authors · 2023-11-22
>
> We thank the reviewer for his positive review and feedback. In the new revision (please refer to our post titled "Summary of changes in updated version" for an overview of the changes made) we have added a paragraph at the beginning of section 6 that answers the reviewer's question on not using procgen in SSED experiments:
>
> As it only permits level instantiation via providing a random seed, Procgen's level generation process is both uncontrollable and unobservable. Considering the seed space to be the level parameter space $\mathbb{X}$ makes the level generation problem trivial as it is only possible to define a single CMDP with $X_C=\mathbb{X}$ and $p(x)=\mathcal{U(\mathbb{X})}$. Instead, we wish to demonstrate SSED's capability in settings where $X_C$ spans a (non-trivial) manifold in $\mathbb{X}$, i.e. only specific parameter semantics will yield levels of the CMDP of interest. As such we pick Minigrid, a partially observable gridworld navigation domain. Minigrid levels can be instantiated via a parameter vector describing the locations, starting states and appearance of the objects in the grid. Despite its simplicity, Minigrid qualifies as a parametrisable simulator capable of instantiating multiple CMDPs.
>
> In short, the simplicity of Minigrid made it a tractable domain for performing experiments and generating datasets of levels for training and evaluation. It was also sufficient for demonstrating SSED's applicability and its advantages over other UED methods. Nevertheless, we will be eager to explore how to best scale up SSED to more complex settings in future work!

---

### Author Response · Authors · 2023-11-22
**Summary of changes in updated version**

We appreciate the reviewers' feedback and apologise for our slow response. We've addressed the reviewers comments in a new revision. Please use the pdfdiff tool for comparing changes by clicking "revisions" below the paper title and then "compare revisions" in the top right corner. The checkboxes can then be used for comparing the latest submission dated "22 Nov 2023, 20:41 Greenwich Mean Time" with the original submission dated "29 Sept 2023, 13:47 British Summer Time".

Below, we outline major changes and improvements, excluding minor modifications like sentence reordering.

Section 1:
- We moved a paragraph about dataset level parameters in practical settings to section 7.
- We’ve added an extra training level to Figure 1's example, since it would not be possible to generalise from only 2 examples with contridactory semantics, as pointed out by mvxM.

Section 2:
- We've included references to prioritised sampling within experience replay buffers and in the multi goal setting, following zMV7's suggestion. We've chosen not to explore these techniques in detail as our work focuses on adaptive level sampling strategies for improving zero-shot generalisation. As such, we chose to study these approaches in terms of their data regularisation capabilities, instead of focusing on their impact on learning efficiency.

Section 3:
- We improved the section's structure by splitting it into four subsections: RL, Contextual MDPs, Generalisation bounds, and Adaptive Level Sampling.
- We introduced an additional Lemma from Bertran et al. (2020) which we use to address one of mvxM's comment in the next section.

Section 4:
- We restructured this section into three subsections for clarity.
- The first subsection establishes the relationship between mutual information and adaptive sampling strategies and clarifies our approach's applicability to both recurrent and non-recurrent architectures.
- The second subsection uses Lemma 3.1 to motivate the decomposition of $V^\pi_i$ and explain how the value prediction objective induces the maximisation of $MI(L,b)$. Conversely $\ell_1$-value loss sampling minimises $MI(L,b)$ in the training data generated, highlighting its role as a data regularisation technique.
- The last subsection describes our key experimental results in Procgen and provides evidence supporting the points made in the first two sections. We also include an additional result measuring the correlation of $MI(L,b)$ with the generalisation gap and with the value loss. This substantiates a claim made in the abstract which was not backed by an experiment in the previous version, as pointed out by mvxM.
- We moved Figure 2 (right) to the appendix in order to make room for the aggregate train scores, which are now reported alongside the test scores and generalisation gap.
- we've included additional analysis and results in appendix B.1 and B.2, including case-studies of two procgen games explaining when adaptive sampling may or may not have a regularisation effect.

Section 5: We clarified that SSED is compatible with any generative technique and is not restricted to a VAE.

Section 6: We clarified why we moved away from Procgen for the SSED experiments, addressing GmmJ's comment.

Section 7: We made our contributions summary more concise and expanded on the potential applicability of SSED in future work.

Section 8: No changes made.

---

### Meta-Review · Area_Chair_v3Tc · 2023-12-06

**Metareview:**

This paper studies the role of mutual information between an RL agent's internal representations and its environment on its generalization performance. Specifically, this paper investigates this question in an environment based on MiniGrid and the Procgen Benchmark. In particular, the paper proposes to adaptively sample environment settings by prioritizing those with lower mutual information with with the agent's representations. They then show that their method results in reductions in generalization gap, and in some cases, improvements in zero-shot transfer compared to baselines like domain randomization, and previous adaptive sampling strategies like PLR.

While the paper presents compelling ideas and an intriguing series of experiments, the meta-reviewer largely concurs with the assessment of Reviewer mvxM, who has provided a clear and thoughtful critique. Two major issues with the current paper is that 1. Mutual information between the agent's representations and the environment serves as the motivating concept behind this work, but is never explicitly measured. The method itself is only conjectured to minimize mutual information in this sense; and 2. Much of the exposition seems overly complex and many terms and notation are not clearly-defined upfront. In particular, the paper mentions "misgeneralisation" several times, but never defines the concept. It is not clear how misgeneralisation here is different from the concept of distribution shift, which is a well-studied problem in ML. Lastly, the paper also introduces the framework of SSED, but it is not clear how this is meaningfully different from the UED setting introduced in previous work.

**Justification For Why Not Higher Score:**

The method at the center of this paper is not clearly performing the motivating idea behind the paper—that of prioritizing levels based on the degree of mutual information. The exposition in the paper is also quite convoluted throughout with some key terms lacking clear definitions.

**Justification For Why Not Lower Score:**

N/A

---

### Decision · Program_Chairs · 2024-01-16

Reject